# Physiological and Transcriptomic Analyses Reveal the Effects of Elevated Root-Zone CO_2_ on the Metabolism of Sugars and Starch in the Roots of Oriental Melon Seedlings

**DOI:** 10.3390/ijms232012537

**Published:** 2022-10-19

**Authors:** Lijia Gao, Wanxin Wang, Chuanqiang Xu, Xintong Han, Yanan Li, Yiling Liu, Hongyan Qi

**Affiliations:** 1College of Horticulture, Shenyang Agricultural University, Shenyang 110866, China; 2Key Laboratory of Protected Horticulture Ministry of Education, Shenyang 110866, China; 3National & Local Joint Engineering Research Center of Northern Horticultural Facilities Design & Application Technology, Shenyang 110866, China

**Keywords:** high root-zone CO_2_, oriental melon seedlings, root growth, sugar and starch metabolism, transcriptomic analysis

## Abstract

Root-zone CO_2_ is a major factor that affects crop growth, development, nutrient uptake, and metabolism. Oriental melon is affected by root-zone gases during growth, the microstructure, sugar and starch contents, enzymatic activities related to sugar and starch metabolism, and gene expression in the roots of oriental melon seedlings were investigated under three root-zone CO_2_ concentrations (CK: 0.2%, T1: 0.4%, T2: 1.1%). Elevated root-zone CO_2_ altered the cellular microstructure, accelerated the accumulation and release of starch grains, disrupted organelle formation, and accelerated root senescence. The sugar and starch contents and metabolic activity in the roots increased within a short duration following treatment. Compared to the control, 232 and 1492 differentially expressed genes (DEGs) were identified on the 6th day of treatment in T1 and T2 plants, respectively. The DEGs were enriched in three metabolic pathways. The majority of genes related to sucrose and starch hydrolysis were upregulated, while the genes related to sucrose metabolism were downregulated. The study revealed that oriental melon seedlings adapt to elevated root-zone CO_2_ stress by adjusting sugar and starch metabolism at the transcriptome level and provides new insights into the molecular mechanism underlying the response to elevated root-zone CO_2_ stress.

## 1. Introduction

CO_2_ concentration in soils is usually determined by soil CO_2_ production and the diffusivity of soils [1], which leads to variations in the CO_2_ concentration at different depths, soil water content, and seasons [2,3]. Owing to the respiration of plant roots and the activities of microorganisms in soils, the concentration of CO_2_ around plant roots is higher than the atmospheric CO_2_ concentration [4]. In particular, the concentration of CO_2_ in soils increases rapidly in agricultural production during waterlogging, soil hardening, and other adverse environmental conditions, which leads to hypoxia in plant roots [5]. The alterations in crop growth and development under elevated root-zone CO_2_ have been widely studied, and the results demonstrated that an increase in rhizospheric CO_2_ concentration can either promote or inhibit crop growth [6,7] depending on a variety of factors, including the type of crop, nutrient uptake, abiotic stresses, concentration of CO_2_, and the periodicity in the variations in CO_2_ concentration [8,9,10]. The majority of these studies have focused on the changes in crop growth; however, the effect of elevated root-zone CO_2_ on the underlying mechanism remains to be elucidated.

A previous study reported elevated root-zone CO_2_ in soils can affect the surrounding plants and soil microorganisms [11], and the CO_2_ that enters plants is utilized during metabolism [12]. The fixation of plant roots in the soil facilitates the uptake and transport of water and mineral nutrients, and is essential for plant metabolism [13,14]. Plants with larger roots have a greater ability to utilize photosynthetic products under elevated root-zone CO_2_, and the photosynthetic products that are allocated to the roots can effectively promote root growth and development [15]. Previous studies on tomato, lettuce, and pepper have demonstrated that elevated root-zone CO_2_ promotes root growth and increases biomass [8,9]. As the products of plant metabolism, sugar and starch are essential factors regulating plant growth, maturation, and senescence [16,17]. These products can also act as signals that influence the cell- and tissue-specific expression of certain genes [18]. It has been reported that the application of high concentrations of CO_2_ with aerated solutions inhibits the root respiration of chickpea and faba bean plants, impedes growth, and subsequently leads to the accumulation of sugar in the root tip cells [19]. However, another study demonstrated that high concentrations of CO_2_ in soils can significantly reduce the starch content in plant roots [20]. These findings indicate that elevated root-zone CO_2_ has varying effects on the contents of sugar and starch in the root tip cells of crops; however, the mechanism underlying these effects remain to be elucidated.

The oriental melon (*Cucumis melo* var. *makuwa* Makino) is an annual crop belonging to the Cucurbitaceae family and is widely cultivated in some eastern Asian countries [21]. The quality and yield of oriental melon are often reduced owing to poor root aeration during cultivation. Rhizospheric hypoxia and elevated root-zone CO_2_ are some of the commonly encountered stresses in agricultural production [5]. Previous studies have demonstrated that the effect of elevated root-zone CO_2_ on crops is more pronounced than that of oxygen depletion, especially in terms of root biomass accumulation [22]. The effects of rhizospheric hypoxia on crop growth and development have been studied extensively [5,19,23]. Previous studies on elevated root-zone CO_2_ have primarily focused on the growth physiology and nutrient uptake of crops; however, there is a scarcity of studies on the alterations in the contents of metabolites, including sugar and starch, especially at the transcriptomic level.

Therefore, this study aimed to investigate the alterations in the accumulation and metabolism of starch and sugar in the root system of oriental melon seedlings under elevated root-zone CO_2_. Three different root-zone concentrations of CO_2_ were selected for treatment, and the previously designed automatic root-zone gas control device was used [24]. The physiological indices of root growth, root cell microstructure, sugar and starch contents, and the activities of enzymes related to sugar and starch metabolism in the roots were determined. Finally, the differences in the physiological indicators of root growth were compared across different time intervals. The time intervals when significant differences were observed were selected for transcriptomic analysis, identification of differentially expressed genes (DEGs), and verification of the key genes with qRT-PCR. The results obtained herein provide novel insights for further studies on the accumulation and metabolism of sugar and starch in plant roots for alleviating the damage resulting from elevated root-zone CO_2_.

## 2. Results

### 2.1. Elevated Root-Zone CO_2_ Affects Root Growth and Vigor

The results demonstrated that the root growth indices tended to increase with the prolongation of treatment duration (Table 1). The total root length, root surface area, root volume, root vigor, and number of root tips increased significantly following exposure to elevated root-zone CO_2_ on the 3rd and 6th days of treatment, compared to those of the control. However, these factors decreased significantly when the duration of treatment was prolonged in the T1 and T2 treatment groups. Elevated root-zone CO_2_ decreased the root surface area on the 6th day of T1 and T2 treatment groups compared to that of the control. In addition, the total root volume of the seedlings in the T1 group increased while that of the T2 seedling decreased significantly on the 9th day of treatment compared to that of the control. The number of root tips with a diameter of 0.0–0.5 mm was significantly higher in T1 seedlings but significantly lower in T2 plants compared to that of the control. Compared to those of the control, all the indices of root growth decreased significantly on the 12th and 15th days of exposure to elevated root-zone CO_2_, and the number of root tips with a diameter of 0.0–0.5 mm decreased by 26.5% and 14.54% in T1 and T2 plants, respectively, while the number of root tips with a diameter of 0.5–2.0 mm decreased by 18.32% and 6.83% in T1 and T2 plants, respectively (*p* < 0.05). The findings revealed that elevated root-zone CO_2_ inhibits root growth with the prolongation of treatment time and consequently regulates root phenotypes via adaptive adjustment of the growth and development of root tips.

### 2.2. Elevated Root-Zone CO_2_ Affected Cellular Ultrastructure of Root Tips

The complete cell diagram depicting the ultrastructure of the root tip cells of oriental melon seedlings in the treatment and control groups is provided in Figure 1. The diagram demonstrates that the cellular structures altered significantly at the beginning of treatment in the T1 and T2 seedlings. Additionally, the mitochondria were in an active state of division at the beginning of treatment, and the number of starch grains gradually increased in the root tip cells compared to that of the control group (Figure 1A—0 d, 3 d, 6 d; Figure 1B—0 d, 3 d, 6 d). The mitochondrial membrane started dissolving and the starch grains were released with the prolongation of treatment time. The number of organelles in the T1 and T2 seedlings reduced significantly compared to that of the CK group on the 9th day of treatment. The accumulation of starch grains in the T1 group increased gradually, while the root tip cells of the T2 plants released the starch grains at the same time. The number of starch grains in T1 cells increased markedly after the 12th day of treatment, and the number of organelles, including mitochondria and Golgi, decreased markedly, while the T2 cells released the majority of starch grains at the same time and the organelles disappeared. Comparison of Figure 1A,B revealed that exposure to elevated root-zone CO_2_ induced the rapid accumulation of starch grains in root tip cells, while higher concentrations of CO_2_ induced the rapid release of starch grains and dissolution of organelles. The findings revealed that exposure to elevated root-zone CO_2_ affected the senescence of root tip cells and the time of release of starch grains, which could be related to sugar and starch metabolism in root tip cells, and can subsequently lead to morphological alterations in roots.

### 2.3. Quality Analysis of Transcriptome Sequencing Data

Quality analysis of the transcriptome sequencing data revealed significant differences among the T1, T2, and CK treatment groups on the 6th day of treatment. We therefore employed the RNA sequencing (RNA-seq) approach for analyzing the samples on the 6th day of treatment. The results of quality analysis of the original data are provided in Table 2. The quality of the library was evaluated and the gene structure level was analyzed. A total of 385.32 million original fragments and 192.66 million paired-end sequence fragment reads were obtained after filtering. The clean reads were mapped to the reference genome of the melon reference genome using the Hierarchical Indexing for Spliced Alignment of Transcripts (HISAT) and Bowtie software. Statistical analyses revealed that 93.07–94.12% of the clean reads were located in the reference melon genome. As the clean reads comprised random cDNA fragments, several random primer sequences showed preference for the 5^′^ end of the reads. We therefore detected the distribution of base types for determining the separation of AT and GC. The GC contents determined in this study are enlisted in Table 2.

### 2.4. Sample Correlation Analysis

A statistical chart depicting the correlation among the nine groups of genes is provided in Figure 2, and the colors indicate the correlation between the samples. Heat map analysis revealed that the correlation coefficients between the treatments at same concentrations of CO_2_ were close to 1, and the correlation between the T1 and T2 samples was also high. The results demonstrated that the gene expression level in the roots of oriental melon seedlings was similar under elevated root-zone CO_2_ stress, and the underlying mechanism was subsequently analyzed.

### 2.5. Screening of DEGs

The DEGs in the different treatment groups were analyzed and two gene libraries were identified by comparison, which included a total of 4171 DEGs, as depicted in Figure 3. A total of 1204 DEGs were significantly differently expressed in the T1 group, of which 738 DEGs were upregulated and 886 were downregulated. The number of DEGs was higher in the T2 plants, in which a total of 2967 genes were differentially expressed, of which 1387 and 1580 DEGs were upregulated and downregulated, respectively. Compared with the T1 group, a total of 398 upregulated DEGs and 559 downregulated DEGs were identified in the T2 group. A total of 232 DEGs were identified in the T1 group, while 1492 genes were significantly differently expressed in the T2 group compared to those of the CK group. The results demonstrated that gene expression varied in the roots of oriental melon seedlings under different root-zone concentrations of CO_2_ on the 6th day, and the number of DEGs in the T2 group was significantly higher than that of the T1 and CK groups. The findings revealed that alterations in the growth and metabolism of the roots of melon seedlings following exposure to elevated root-zone CO_2_ could be attributed to differential gene expression, which alters the metabolism of root cells at the transcriptomic level and consequently affects the activity of the entire plant.

### 2.6. Kyoto Encyclopedia of Genes and Genomes (KEGG) Pathway Analysis of DEGs

The significantly enriched KEGG pathways of the identified DEGs and the enrichment levels of the DEGs were determined by KEGG pathway analysis (Figure 4). A total of 208 and 464 DEGs of T1 and T2 plants, respectively, identified by comparison of the T1/CK, T2/CK, and T1/T2 groups were annotated to 82 and 107 metabolic pathways, respectively, by KEGG pathway analysis. The 170 genes in the T1/T2 groups were significantly enriched in 74 KEGG metabolic pathways. The right panel in Figure 4 depicts the 20 most significantly enriched metabolic pathways. Analysis of the significantly enriched metabolic pathways in both the treatment groups revealed that the T1/CK, T2/CK, and T1/T2 groups were significantly enriched in three pathways, namely “phenyl propanoid biosynthesis”, “plant hormone signal transduction”, and “starch and sucrose metabolism”. The “amino acid biosynthesis” pathway in T1/T2 and T2/CK was also significantly enriched. The former two pathways, namely “phenyl propanoid biosynthesis” and “plant hormone signal transduction”, are associated with plant resistance to stress. As the significantly enriched pathways are related to starch and sucrose metabolism, the findings indicated that elevated root-zone CO_2_ stress was the primary reason underlying the alterations in the contents of sugar and starch in the roots at the gene level. Apart from the “phenyl propanoid biosynthesis” and “starch and sucrose metabolism” pathways, the DEGs in the T1 group were also enriched in the “linoleic acid metabolism” (KO00591) and “α-linolenic acid metabolism” (KO00592) pathways, while the DEGs in the T2 group were enriched in the “L-phenylalanine metabolism” (KO00360) and “flavonoid biosynthesis” (KO00941) pathways. These findings indicated that the seedlings of oriental melon could also respond to elevated root-zone CO_2_ stress via the aforementioned metabolic pathways. We therefore speculated that the accumulation of starch grains in root tip cells could be related to the post-transcriptional pathways that were enriched in the DEGs.

### 2.7. DEGs Involved in Sugar and Starch Metabolism

A total of eight genes related to starch and glucose metabolism were identified by transcriptome analysis (Table 3), including *CmBAM* (MELO3C006362.2), *CmTS* (MELO3C005751.2), catalytic subunit of SNF1-related protein kinase (SNFK1; *α-KIN1*, designated as *CmSNFK1* (MELO3C026210.2)), *CmTPS* (MELO3C025673.2), *CmSS* (MELO3C017942.2), *CmAMY* (MELO3C017002.2), *CmNI* (MELO3C006727.2) and *CmSPS* (MELO3C020357.2). *CmBAM*, *CmTS*, and *CmSNFK1* were upregulated in the T1 and T2 seedlings, while *CmAMY* was downregulated in the T1 and T2 plants, and the expression of *CmTPS*, *CmSS*, and *CmNI* was upregulated in the T2 group. The expression of *CmSPS* was downregulated in T2 plants compared to that of the T1 seedlings. The results demonstrated that the transcription levels of the genes related to starch and sucrose hydrolysis were generally upregulated, while the transcription levels of the genes related to sucrose synthesis were downregulated following exposure to elevated root-zone CO_2_. The findings also indicated that elevated root-zone CO_2_ stress regulates sugar and starch metabolism in the roots of oriental melon seedlings at the transcriptome level for adapting to adverse conditions. Additionally, the upregulation of genes related to trehalose synthesis and the *CmSNFK1* gene revealed that the plant could adapt to adverse conditions by activating the sugar signaling pathways and regulating the expression of genes related to the metabolism of sugar and starch.

### 2.8. Elevated Root-Zone CO_2_ Affected Contents of Soluble Sugars and Starch in Roots

Analyses of the ultrastructure of root tip cells and regulation of the identified DEGs at the transcriptional level revealed that elevated root-zone CO_2_ induced the rapid formation of starch grains in root tip cells. We therefore investigated the contents of starch and soluble sugars in the root tip cells. The contents of soluble sugars and starch in the roots of oriental melon seedlings initially increased and decreased later on with the prolongation of treatment duration, as depicted in Figure 5. The contents of soluble sugars and starch increased significantly at first in both the T1 and T2 treatment groups compared to those of the control group. Compared to those of the control group, the contents of soluble sugar and starch were significantly increased in the T1 group but significantly reduced in the T2 group on the 9th day of treatment. The contents of soluble sugars and starch in the T1 and T2 groups decreased significantly after the 12th day of treatment compared to those of the control group. The findings revealed that the inhibitory effect of elevated root-zone CO_2_ on the contents of soluble sugars and starch was enhanced as the concentration of CO_2_ was increased and was consistent with the alterations in root phenotype.

### 2.9. Elevated Root-Zone CO_2_ Affected the Contents of Different Sugars in Roots

Sucrose, raffinose, and stachyose are involved in long-distance transport in Cucurbitaceae. Stachyose is the main photosynthetic product and transport substance in oriental melon, while sucrose and raffinose also account for a certain proportion. We therefore determined the contents of several monosaccharides in the seedlings of oriental melon. As depicted in Figure 6, the content of fructose following exposure to elevated root-zone CO_2_ decreased initially and increased later on with the prolongation of treatment duration compared to that of the control. The contents of fructose, trehalose, stachyose, and raffinose in the T1 and T2 groups decreased significantly on the 9th day of treatment compared to those of the control but were significantly higher than those of the control after the 12th day of treatment. However, the glucose contents of the T1 and T2 groups reached a peak on the 6th day of treatment and were significantly higher than that of the control, but decreased gradually thereafter. The findings confirmed that glucose plays a vital role in the alterations in the total soluble sugar content. The sucrose content increased on the 3rd day of treatment and decreased thereafter, but remained unaltered as the duration of treatment was prolonged or the concentration of CO_2_ was increased. The results indicated that sucrose synthesis and transportation were inhibited in root tip cells after the 6th day of treatment. The findings also revealed that elevated root-zone CO_2_ initially increased monosaccharide transportation from source to sink and the loading demands of oriental melon.

### 2.10. Elevated Root-Zone CO_2_ Affected Activities of Enzymes Related to Sugar and Starch Metabolism in Roots

The acid invertase (AI) activity of the T1 and T2 groups increased from the 6th and 9th days, respectively, of exposure to elevated root-zone CO_2_ compared to that of the control (Figure 7). However, the activity of neutral invertase (NI) in the roots of the both the treatment groups was significantly higher than that of the control as the duration of treatment was prolonged. The activity of sucrose phosphate synthase (SPS) varied under different concentrations of CO_2_, and peaked on the 9th and 6th days of treatment in the T1 and T2 groups, respectively. The activity of sucrose synthase in the decomposition direction (SSI) increased gradually and significantly as the concentration of CO_2_ was increased. The activity of α-amylase was significantly lower than that of the control on the 6th day of treatment but gradually increased thereafter with the prolongation of treatment duration. The activity of β-amylase increased significantly on the 6th and 9th days of treatment in the T1 and T2 groups, respectively, and remained stable thereafter. The activity of trehalose synthase (SSI) increased initially but decreased later on, which was consistent with the changes in the activity of SPS. The enzymatic activities in both the treatment groups were weaker than those of the control group on the 15th day of treatment, while the activity of trehalose phosphate synthesis (TPS) gradually increased on the whole. The alterations in the enzymatic activities were consistent with the changes in the contents of soluble sugars and starch, which indicated that elevated root-zone CO_2_ affected the metabolism of sugars and starch in the roots at the gene level.

### 2.11. qRT-PCR Analysis of Major Genes Related to Sugar and Starch Metabolism

The expression of genes related to sugar metabolism in the roots was further determined by analyzing the expression of related genes in combination with transcriptome analysis during treatment (Figure 8). qRT-PCR analysis of gene expression in the roots of oriental melon seedlings under three different concentrations of root-zone CO_2_ revealed that the relative expression levels of the *CmSPS* genes, which are related to sucrose synthesis, decreased significantly on the 6th and 9th days of treatment in the T1 and T2 groups, respectively, compared to those of the control. However, the expression of *CmSS* and *CmNI* genes, which are related to the decomposition of sucrose, was upregulated on the 6th day after treatment with high concentrations of CO_2_, and the relative expression levels increased with an increase in the concentration of CO_2_. The expression of *CmAMY*, which is related to the hydrolysis of starch, was high at elevated root-zone CO_2_ on the 3rd day of treatment but decreased on the 6th day, while the expression of *CmBAM* increased significantly from the 6th day of treatment. The expression levels of *CmSNFK1*, an SNRK1 gene closely related to the T6P signaling pathway in response to adverse conditions, also increased significantly on the 6th day of treatment. The findings revealed that the contents of sucrose in the roots of oriental melon seedlings were sufficient during the first three days of treatment in the T1, T2, and CK groups. The active hydrolysis of the sucrose and starch in roots increases the content of soluble sugars, and the accumulation of the hydrolysates triggers the synthesis of T6P, which subsequently inhibits the SnRK protein kinase. With the consumption of sucrose, T6P was converted to trehalose under the activity of *CmTS*, which activated the T6P-SnRK protein kinase signaling pathway. This in turn enhanced the expression of *CmSNFK1*, which subsequently accelerated the transport of carbohydrates in plants during days 9–15 of treatment. The activities of *CmSPS* and *CmTPS* were subsequently inhibited, which established a new carbohydrate balance in the roots.

## 3. Discussion

### 3.1. Elevated Root-Zone CO_2_ Affected Root Growth and Accumulation of Sugars and Starch

The development of roots is an essential agronomic trait in plant growth [25], and a strong root system is the key to plant productivity. The rich epidermal structure of the roots facilitates communication between plants and the soil environment, which provides sufficient water and nutrients for plants [26]. The species of plants, composition of soils, and nutrients greatly influence the spatial structure of the roots, including the number and length of lateral roots [27]. In this study, we observed that elevated root-zone CO_2_ promoted the development and growth of roots of oriental melon seedlings in the first 6 days of treatment, indicated by an increase in total root length, number of root tips, larger root surface area, increased accumulation of root biomass, and increased root activity (Table 1). These findings were consistent with the results of previous studies which reported that elevated root-zone CO_2_ promotes the growth and development of the roots of lettuce in tropical regions [10]. However, the results of this study demonstrated that elevated root-zone CO_2_ decreased root growth in the seedlings or oriental melon with the prolongation of treatment duration, while the degree of inhibition was positively correlated with the concentration of CO_2_. This phenomenon improved the root vigor and enhanced the metabolism and absorptive capacity in the short term, but they were significantly inhibited thereafter. This was probably attributed to the fact that elevated root-zone CO_2_ significantly alters the pH of root cells with the prolongation of treatment duration, which subsequently suppresses the membrane function of root cells and disrupts osmotic regulation and balance [28].

Analysis of the ultrastructure of root tip cells revealed that the cell membranes, organelles, and cell walls gradually dissolved under elevated root-zone CO_2_ (Figure 1). The accumulation of starch grains and mitochondrial division increased rapidly on the 3rd day of treatment with 1.1% root-zone CO_2_ in the T2 group. The starch grains were rapidly released on the 6th day of treatment while the mitochondrial membrane began dissolving and breaking up on the 9th day of treatment in the T2 group. In addition, large cavities were observed on the cell surface, and cellular physiological functions were lost on the 15th day of treatment. However, the accumulation of starch grains was observed on the 6th day of treatment with 0.4% root-zone CO_2_ in the T1 group, and the mitochondria and starch grains were intact on the 9th day of treatment. Cytolysis was observed in the T1 treatment group on the 15th day of treatment, but the extent of cytolysis was lower than that of T2 plants. Previous studies have demonstrated that the synthesis and degradation of starch are regulated by redox signals [29], and the synthesis of starch is closely related to the mitochondrial metabolism in non-photosynthetic tissues [30]. These findings indicated that the formation of starch grains in root tip cells under elevated root-zone CO_2_ was related to mitochondrial activity in this study. In addition, the results of transcriptome analysis revealed that the expression of genes related to sugar and starch metabolism was upregulated, and the genes were enriched in several pathways related to sugar metabolism. This further indicated that the differences in the root growth of oriental melon seedlings under elevated root-zone CO_2_ could be attributed to the alterations in the levels of starch metabolism. The results demonstrated that although the root growth phenotype was not inhibited by 1.1% CO_2_, the interior of the root tip cells exhibited signs of senescence, which was not conducive to the growth and development of melon seedlings. However, the root tip cells of the T1 group, which was treated with 0.4% CO_2_, remained active on the 9th day of treatment.

### 3.2. Elevated Root-Zone CO_2_ Affected the Metabolism of Sugar and Starch in the Roots

The development, stress response, and yield formation of plants are influenced by sugar metabolism. The metabolism of sugars primarily promotes plant growth and the synthesis of essential compounds, including proteins, cellulose, and starch, by generating a generating a series of sugars as metabolites, and acts as signal molecules to regulate transcription factors and other gene expression [31,32,33]. Sugar signaling pathways interact with stress pathways to regulate plant metabolism. Sugars indirectly regulate carbohydrate metabolism under abiotic stress, which contributes to plant growth and development [34,35,36]. Several environmental factors, including drought, cold, and salinity, have been shown to alter carbohydrate metabolism [37,38,39]. The results of this study demonstrated that alterations in the contents of soluble sugars and starch in the roots are attributed to the adaptation of the basic physiological indices of roots at different root-zone CO_2_ concentrations (Figure 5). In this study, the content of soluble sugars increased with an increase in the concentration of CO_2_ from days 0 to 6 for adapting to the elevated root-zone CO_2_, which was consistent with the increased tolerance of plants under low oxygen concentrations [30,40,41]. However, the starch content increased with an increase in the concentration of CO_2_ from days 0 to 9 (Figure 5), which could be attributed to the fact that the carbon source in the root tip cells were sufficient and induced the absorption of CO_2_ by the roots, and the malic acid content in the roots increased temporarily, which led to the accumulation of starch [42]. The findings could also be explained by the compensatory response to elevated soil CO_2_ stress. In a previous study, Lake et al. proposed a root-to-leaf signaling mechanism under elevated root-zone CO_2_ stress, which explained that the roots of plants are stimulated to produce hormones under elevated root-zone CO_2_ stress, and a signal is transmitted to the leaves for inducing stomatal closure. As a compensatory mechanism to root-zone CO_2_ stress, the starch in the root system can be hydrolyzed into soluble sugars [43]. Increased exposure to root-zone CO_2_ resulting from the prolongation of treatment duration significantly reduces the pH of the root sap, which negatively affects root tip cells and markedly reduces the starch content. However, previous studies have demonstrated that the expression of genes related to the conversion of starch and sucrose in the source-sink relationship is affected by sugar signaling pathways. The T6P sugar signaling pathway plays an important role in promoting AGPase redox activation in response to the conversion of sucrose [44,45]. In this study, the activity of TPS, the T6P synthesis ability, and the upregulation of starch synthesis decreased after the 9th day of treatment with elevated root-zone CO_2_, while the contents of α-hydrolase and β-amylase in the catabolic direction increased compared to those of the control, which consequently decreased the accumulation of starch (Figure 7).

The enhanced activity of AI in melon can result in the formation of hexoses, which serve as a carbon source for the rapid growth of plant tissues [46]. Glucose and fructose are the main hexose sugars in the rapidly expanding young tissues of melon, which often have very high invertase activity [47,48]. The results of this study demonstrated that the activity of NI increased significantly under elevated root-zone CO_2_, and the increase in the activity of NI was more significantly pronounced with the prolongation of treatment time and an increase in the concentration of CO_2_. The activity of SS generally increased, but it decreased with the prolongation of treatment time (Figure 7). This finding was consistent with the results of previous studies on rice [49], wheat [50], and corn [51]. In this study, SS played an important role in root growth on days 0–6 of treatment. The activity of SS increased with an increase in the concentration of CO_2_. The activity of SPS was significantly lower than that of the control after the 6th day of treatment and decreased with an increase in the concentration of CO_2_; however, the activity of SPS in the T2 treatment group increased significantly compared to that of the control on the 3rd day of treatment (Figure 4). As depicted in Figure 5, the sucrose content increased to a peak and decreased later on but remained steady over the remaining duration of treatment and at different CO_2_ concentrations. The contents of fructose and glucose increased simultaneously at first but declined later on (Figure 6). The findings indicated that the direction of sucrose decomposition in root tip cells gradually became more pronounced compared to sucrose synthesis after the 3rd day of treatment, which consequently increased the sucrose consumption and decreased the sucrose accumulation, and caused a gradual reduction in the content of soluble sugars. On the other hand, the contents of T6P and TS initially increased and then decreased later on. The contents of T6P and TS increased significantly on the 6th day of treatment with 1.1% CO_2_ while the content of trehalose increased significantly on the 6th day of treatment as the concentration of CO_2_ increased. The findings revealed that the trehalose synthesis signal was stimulated in plants under elevated root-zone CO_2_ over days 0–6 of treatment, which resulted in the accumulation of trehalose. The observation was closely related to the alterations in sucrose content and was consistent with the results of a study on grape under cold stress [52]. Another study on cottonseed reported that the contents of sugar and stachyose generally increases from the 6th day of treatment, and the alterations in the contents of sugars and stachyose were similar at different treatment concentrations. The contents of sugars and stachyose increased significantly after the 9th day of treatment compared to those of the control, which could indicate that elevated root-zone CO_2_ increased the transportation and loading demand of melon from source to sink from the 6th day of treatment, thereby forming a new metabolic balance. Elevated root-zone CO_2_ induced a series of changes in the sugar metabolism of roots and thereby directly affected the material metabolism of the roots. This phenomenon provided a physiological and biochemical basis for the morphological and structural adaptation of the roots of oriental melon seedlings.

The findings of this study revealed that the metabolism of sugars and starch in the roots of oriental melon seedlings undergoes a series of changes under elevated root-zone CO_2_, which subsequently promotes the accumulation and transformation of carbohydrates in the short term. Alterations in sugar metabolism adjust the cellular balance of carbon and water metabolism and aid in stress adaptation. However, the balance between the supply and demand of carbohydrates is disrupted after long-term treatment, including a reduction in photosynthetic capacity, cellular pH, and other adverse factors, which subsequently inhibits plant growth. The results of transcriptome analysis (Figure 4) and KEGG pathway analysis revealed that the DEGs were significantly enriched in three KEGG pathways, including “sugar and starch metabolism”. This pathway is involved in the defense response of melon seedlings and was the most significantly enriched metabolic pathways, which indicated that the metabolism of sugars and starch plays an important role in the defense response of oriental melon seedlings. Quantitative analysis of the relevant genes identified by transcriptome analysis revealed that all the genes related to sucrose decomposition were upregulated from days 0 to 6 (Figure 8). The elevation of root-zone CO_2_ upregulated the expression of the related genes to varying degrees. Of these, the expression of *CmNI*, *CmAI*, and *CmSSI* was significantly altered on the 3rd day of treatment, while the overall expression levels of *CmSPS*, *CmTPS*, and *CmTS* were first upregulated and then downregulated. The expression of *CmSPS* was significantly downregulated in the T1 and T2 treatment groups after the 9th day of treatment compared to that of the control group, and the expression of *CmSPS* was downregulated at higher concentrations of CO_2_. The expression levels of *CmTPS* and *CmTS* were significantly higher in the T1 and T2 groups than those of the control group on the 6th day of treatment, while the expression levels decreased after the 9th day of treatment. Analysis of gene expression in the roots of melon seedlings by qRT-PCR under the three different root-zone concentrations of CO_2_ revealed that the variations in gene expression were fundamentally similar to the alterations in enzymatic activity and was consistent with the results of transcriptome analysis. The activity of SPS is regulated via phosphorylation. SPS and invertase cooperatively control the long-distance transportation and metabolism of sucrose in the sink tissues [53]. Previous reports have suggested that the highly conserved SNFK1 is involved in the redox regulation of AGPase and the signal transduction pathways that mediate the synthesis of starch from sugar [36]. It has also been reported that trehalose and its precursor T6P regulate plant metabolism and gene expression. In addition, the synthesis, decomposition, and regulation of trehalose and T6P play an important signaling role in regulating plant growth, development, and adapting to stress [54]. In this study, the expression of *CmSNFK1*, which is closely related to T6P signaling in response to adverse conditions, increased significantly on the 6th day of treatment. This finding indicated that the melon seedlings were rich in sucrose, which served as a sufficient carbon source under elevated root-zone CO_2_ from days 0 to 3 of treatment. *CmNI*, *CmAI*, and *CmSS* actively regulated the activities of enzymes related to the hydrolysis of sucrose and starch, and increased the content of soluble sugars. The accumulation of their hydrolysis products triggered the synthesis of T6P, which in turn inhibited the SnRK protein kinase and promoted the consumption and utilization of carbon sources by the roots to regulate the carbon and nitrogen imbalance caused by elevated root-zone CO_2_ stress. The findings were consistent with those of previous studies which reported that alterations in the levels of T6P in plants affect the response of *Arabidopsis* sp. to sugar, which indicated that T6P plays an important role in regulating sugar metabolism and can also alter and regulate plant growth and carbon utilization [55]. The utilization of the carbon content in roots decreases following the destruction of root tip cells by the acidic environment, which subsequently activates the T6P-SnRK protein kinase signaling pathway and increases the expression of *CmSNFK1*. In this study, the carbon consumption of the root system was gradually reduced, the activity of SPS, TPS, NR, and other enzymes increased, and the source-sink transportation of carbon was accelerated for establishing a new carbon and nitrogen balance.

## 4. Materials and Methods

### 4.1. Plant Material and Growth Conditions

The experiments were conducted in a solar greenhouse in the scientific research base of Shenyang Agricultural University, Shenyang, Liaoning, China. The Yumeiren cultivar of oriental melon (Hengyuan Seed Industry, Changchun, Jilin, China) was selected as the plant material, and an aeroponics system was used for cultivation. When the seedlings grow three leaves, the healthy seedlings with uniform growth that were free of pests and diseases were selected for planting. The substrate was washed from the roots of the seedlings and the seedlings were transplanted to the planting holes of the cultivation bed containing cotton substrate.

The seedlings were treated with an automatic root-zone CO_2_ concentration treatment system based on the root-zone CO_2_ treatment method by Chen et al., 2020. (Figure 9) [24]. The root-zone CO_2_ cultivation and treatment system mainly comprised a CO_2_-air mixing system, a CO_2_ concentration monitoring system, and an aeroponic cultivation system. Three weeks after transplantation, three different root-zone CO_2_ concentrations (0.2%, 0.4% and 1.1%) were supplied to separate bins. Continuous elevated CO_2_ treatment lasted for 15 days. The CO_2_ concentrations of 0.2%, 0.4% and 1.1% were controlled using premixed CO_2_–air mixtures from compressed air and high-purity CO_2_ cylinders. The gas was mixed by adjusting the intake of CO_2_ and air and maintained the CO_2_ concentration at the preset CO_2_ level. The gas mixture was introduced through a pipe inserted near the root. There were three bins for each root-zone CO_2_ concentration treatment. The root-zone CO_2_ concentration of the seedlings could be monitored by CO_2_ and an oxygen sensor located in the cultivation bed. The computer data-acquisition module collected the data from CO_2_ and the oxygen sensor in the cultivation bed and then transmitted the data to the computer control module to ensure the CO_2_ concentration in the cultivation bed was at the pre-set CO_2_ level. The computer simultaneously carried out the real-time monitoring, data recording, and storage. All experiments were performed in a greenhouse kept at the same temperature.

### 4.2. Measurement of Plant Growth Indices

Samples were collected from day 0, and on every 3rd day until the 15th day of treatment. The plants at the same stage of growth in the different groups (CK, T1, and T2) were selected, and the root volume, activity, and morphology were determined. The data obtained from three independent experiments were averaged, and three biological replicates were considered. The root volume was determined using the drainage method. The root vigor was measured by the triphenyltetrazolium chloride (TTC) method.

### 4.3. Analysis of Root Morphology

The intact roots were scanned using a Wanshen LA-S scanner (Hangzhou, China), and a series plant image analyzer, and quantitative analysis was achieved with the matching system.

### 4.4. Observation of Ultrastructure of Root Tip Cells

The samples were prepared as previously described by Chen et al. [25] and observed using a JEX-100CX II transmission electron microscope (JEOL Ltd., Tokyo, Japan).

### 4.5. Determination of Sugar and Starch Contents

The contents of soluble sugars and starch were determined by the anthranilic sulfuric acid method with 0.1 g of the root samples. The contents of sugars were determined by high performance liquid chromatography (HPLC) using 1.0 g of the root samples. Briefly, 5–6 mL of 80% ethanol was added to the samples, and the samples were immersed in a water bath at 80 °C for 1 h. The process was repeated thrice and the extracts were combined. The extracts were allowed to cool and subsequently centrifuged at 8000 rpm for 10 min. The obtained supernatant was poured into an evaporating pan and dried in a water bath at 80–90 °C. The evaporated sugars were dissolved in 1 mL of ultrapure water. The solution was filtered using a 0.22 μm water filtration membrane, and the supernatant was added to the sample bottle. Standard samples of various sugars (Shanghai Yuanye Biotechnology Co., Ltd. Shanghai, China) were prepared at concentrations of 0.2, 0.4, 0.8, 1, 2, and 4 mg/mL. HPLC was performed using a Waters 600E HPLC (Waters, Milford, CT, USA)system attached with a Waters 2410 (Waters, Milford, CT, USA) Differential Refractometer. A carbohydrate column was used and the temperature of the column was set to 30 °C. The mobile phase comprised acetonitrile and ultrapure water at a ratio of 70% to 30%, and the Waters Millennium Software (version 32) was used for process control and data analysis.

### 4.6. Determination of Activities of Enzymes Related to Sugar and Starch Metabolism

The activities of neutral invertase (NI) [56], sucrose synthase in the decomposition direction (SSI) [57], α-amylase, and β-amylase were determined using a Solarbio Kit (Beijing Solebo Technology Co., Ltd., Beijing, China). The activities of sucrose phosphate synthase (SPS), trehalose synthetase (TS), and trehalose phosphate synthetase (TPS) were detected using a plant sucrose phosphate synthetase, trehalose synthetase, and trehalose hexaphosphate synthetase ELISA Kit, respectively (Jiangsu Boshen Biotechnology Co., Ltd., Taixing, China).

### 4.7. Analysis of Transcriptome Sequencing Data

The alterations in the physiological indices and sugar contents in the T1 and T2 groups exhibited an opposite trend to those of the CK group until the 6th day of treatment. The CK, T1, and T2 groups were selected for subsequent analyses, and 2 cm of the root tips on the 6th day of treatment were selected for RNA-seq analysis. Three biological replicates were performed for each treatment (for a total of 9 samples). The three replicates for each sample were designated as CK-1, CK-2, CK-3, T1, and T2 as the same as above.

For RNA-seq analysis, the total RNA was first extracted using an NEBNext Ultra^TM^ RNA Library Extraction Kit (NEB, Ipswich, MA, USA) and the integrity of the RNA was detected using an RNA Nano 6000 Analysis Kit (Agilent Technologies, Santa Clara, CA, USA) and an Agilent Bioanalyzer 2100 System (Agilent Technologies, Santa Clara, CA, USA). The RNA samples were subsequently clustered and sequenced, and the M-MuLV reverse transcriptase and random primers were used for synthesis. Reverse transcription was performed using DNA polymerase I and RNase H. The gene fragments were amplified using Universal and Index(X) primers and Phusion DNA polymerase, and cDNA used as the template. The quality and quantity of the library were verified after quality testing, and the raw data were filtered to obtain the clean data.

The HISAT2 tool was used for comparing the genome sequences of melon with the obtained mapped data for further analysis, and the DEGs were subsequently identified. The number of mapped reads and transcript lengths of the samples were normalized using String Tie (version 1.3.3). The results were summarized with Gffcompare for determining the gene expression levels based on the values of FPKM.

### 4.8. Analysis of Expression of Genes Related to Sugar and Starch Metabolism

The accuracy of the sequencing results and the regularity of expression were verified by qRT-PCR. The primer sequences are enlisted in Table 4 (designed by Beijing SaiBaiSheng Technology Co., Ltd., Beijing, China). The cDNA was synthesized using a qRT-PCR reverse transcription kit (provided by Kangwei Century Biotechnology Co., Ltd., Beijing, China). Amplification was performed using the Trans Start Top Green qPCR Super Mix of Bio-Rad CFX96 (Bio-Rad, Hercules, CA, USA). The conditions of PCR were as follows: initial denaturation at 94 °C for 2 min, followed by 45 cycles of denaturation at 94 °C for 5 s, extension at 60 °C for 15 s, and annealing at 72 °C for 10 s. The expression level of the product was calculated using the 2^−^^∆∆CT^ method (*p* < 0.05)

### 4.9. Data Processing and Statistical Analysis

The data were presented as the mean ± Standard Error (SE). Analysis of variance (ANOVA) and significance analysis were performed using DPS software, version 17.10. Significance analysis was performed by Duncan’s multiple range test at *p* < 0.05 and Origin version 9.0 (OriginLab Corporation, Northampton, MA, USA). Microsoft Excel 2010 was used for mapping the data and statistical analysis.

## 5. Conclusions

In this study, the effect of elevated root-zone CO_2_ on the seedlings of oriental melon cultivated in an aeroponics system was investigated. The findings revealed that elevated root-zone CO_2_ promoted the growth and vigor of roots of melon seedlings within a short time, increased the contents of sugar and starch in the roots, and improved the activity of the metabolic enzymes. However, the values of all the aforementioned indicators decreased with the prolongation of treatment time, and the growth and metabolism of the seedlings were subsequently inhibited. Transcriptome analysis revealed that the expression of genes related to sugar and starch metabolism declined significantly under elevated root-zone CO_2_ over time. The inhibition of gene expression could be attributed to the fact that elevated root-zone CO_2_ stress affected the carbohydrate metabolism of oriental melon at the gene level for triggering the defense stress response of seedlings and accelerating the aging of root tip cells, which subsequently altered the accumulation and metabolism of sugars and starch in the roots.

## Figures and Tables

**Figure 1 ijms-23-12537-f001:**
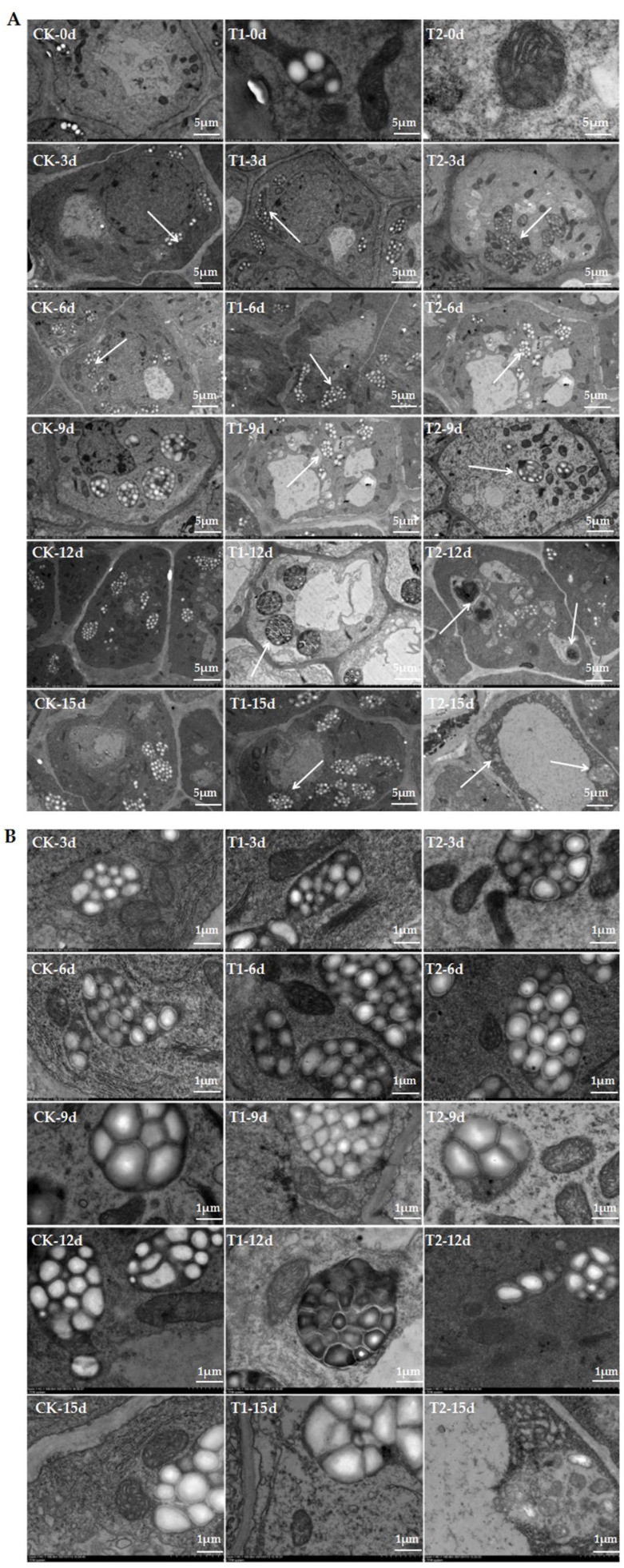
Elevated root-zone CO_2_ concentration on cellular ultrastructure of oriental melon seedling roots (**A**). Ultrastructure changes in root tips after treatment of high root-zone CO_2_. The magnification is 2.0 K. Scale bar: 5.0 µm. The white arrows in CK-3d, T1-3d and T2-3d indicated that the start of the formation of starch grains within the cells. The white arrows in CK-6d, T1-6d and T2-6d indicated that the content of starch grains under T2 treatment was significantly higher than that of CK and T1. The white arrows in T1-9d and T2-9d indicated that the number of organelles began to dissolve and disappear. The white arrows in T1-12d and T2-12d indicated that there were fewer organelles in T2 than in T1 on the 12th day. The white arrows in T1-15d and T2-15d indicated that the organelles in T2 were completely dissolved compared with T1. Day 0 means no treatment. On the 3rd day, it was observed that the number of mitochondria was significantly increased, and the number of starch grains under T1 was larger than that of the T2 treatment. On the 6th day, it was observed that the starch grains under the CK group (control check, elevated root-zone CO_2_ concentration 0.2%) began to accumulate, accompanied by the increase in mitochondria, while the amyloid membrane of T2 began to rupture and dissolve, then the starch grains were released. On the 9th day, starch grains treated with the increase in root-zone CO_2_ concentration began to accumulate in the inner cells of the roots, the number of mitochondria was significantly reduced and cavities appeared, while the starch grains under T2 treatment of the amyloplasts were mostly released. On the 12th day of treatment, the mitochondria in the apical cells were deformed, the number of inner cristae decreased and the edges were blurred. On the 15th day, only a small number of recognizable organelles remained in the apical cells under T1 treatment, the starch grains of T2 root tip cells were released, and starch grains of the CK group began to be released but mitochondria could still be found. (**B**) Ultrastructure changes in root tips after elevated root-zone CO_2_ treatment for 3, 6, 9, 12 and 15 days. The magnification is 1.0 K. Scale bar: 1.0 µm.

**Figure 2 ijms-23-12537-f002:**
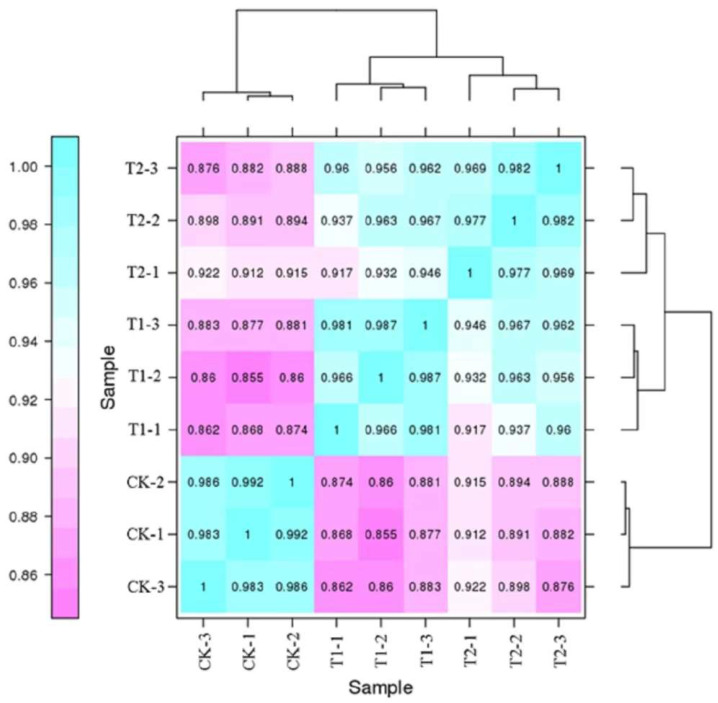
Heat map of gene expression correlation among processed samples. Different colors represent the magnitude of the values: closer to blue means higher correlation between samples, and closer to purple means lower correlation between samples. CK-1, CK-2 and CK-3 represent three sample replicates of the control group, T1 and T2 as above.

**Figure 3 ijms-23-12537-f003:**
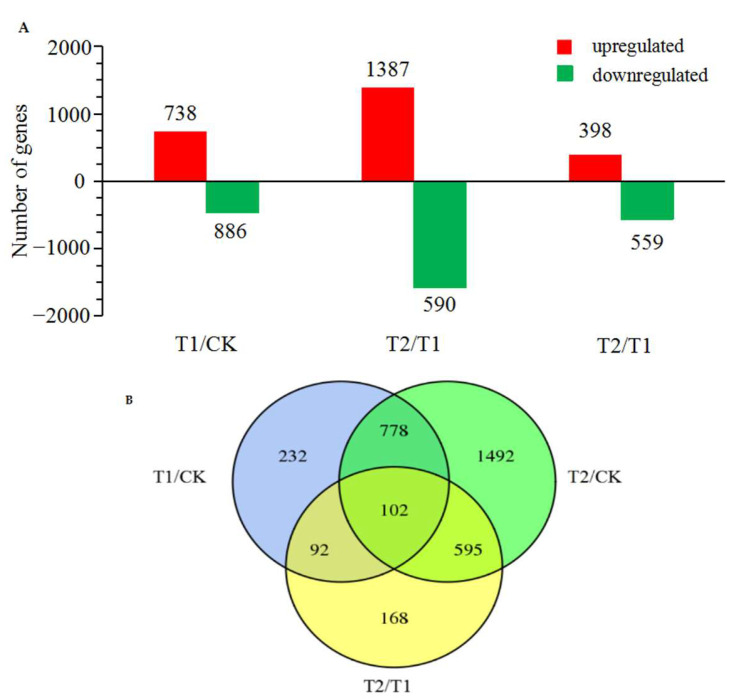
Distribution of the number of up-regulated and down-regulated differentially expressed genes (DEGs) between treatments (**A**). Red indicates upregulation, green indicates downregulation. (**B**) DEG distribution of samples in T1/CK, T2/CK and T2/T1. The criteria of|Log2FoldChange| ≥ 1 and false-discovery rate (FDR) ≤ 0.05 were used to indicate the significance of differentially expressed genes (DEGs).

**Figure 4 ijms-23-12537-f004:**
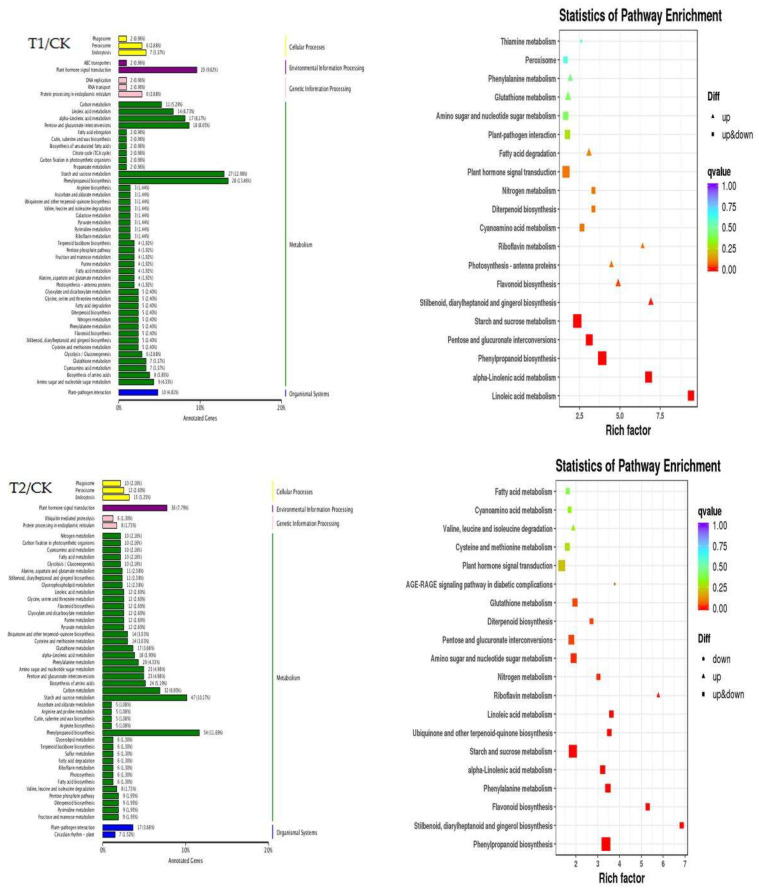
Functional classification and enrichment analysis of DEG in KEGG (**left**). The x-axis represents the number of annotated DEGs under each pathway and its proportion to the total number of genes; The y-axis represents the name of the KEGG pathway, namely cellular processes, environmental information processing, genetic information processing, metabolism and biological systems. Enrichment analysis of differential gene KEGG pathway (**right**). The *x*-axis represents the enrichment factor and the *y*-axis represents the KEGG pathway. The color indicates the significant level of DEGs, and the size of the dot indicates the number of DEGs enrichment.

**Figure 5 ijms-23-12537-f005:**
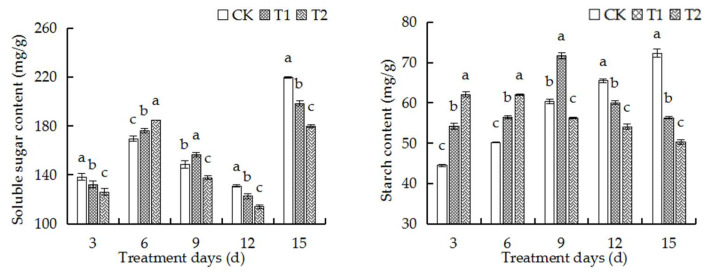
Effects of high root-zone CO_2_ on soluble sugar content and starch content of oriental melon seedling roots. Triplicate biological replicates were performed. The error bars indicate the standard error (SE) of the mean. Significant differences are represented by different lowercase letters (a, b and c represent *p* < 0.05).

**Figure 6 ijms-23-12537-f006:**
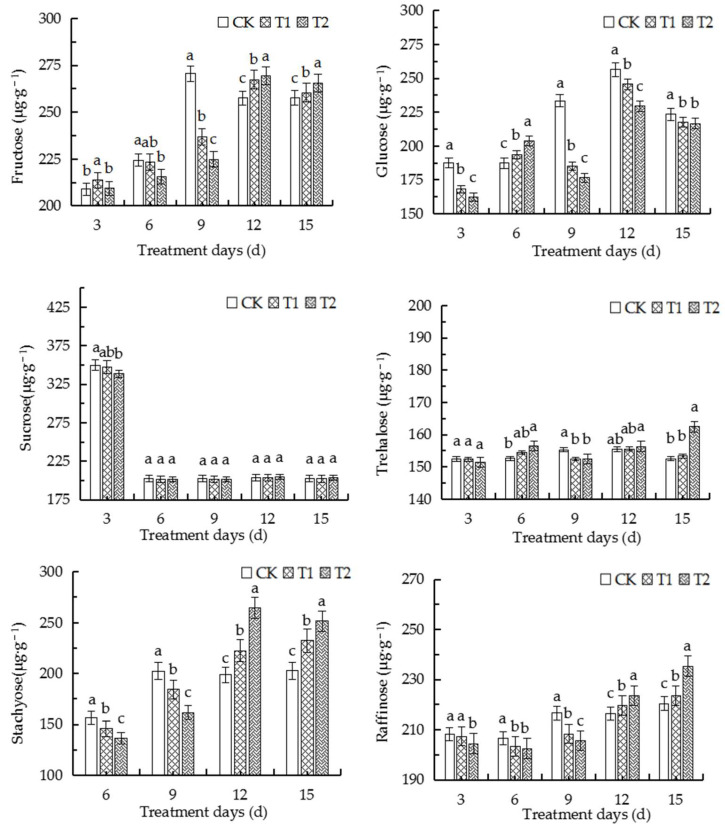
Alterations in the sugar content in the roots of oriental melon seedlings, including fructose, glucose, sucrose, trehalose, stachyose, and raffinose, following treatment with high root-zone CO_2_. The biological replicates were performed in triplicate. The error bars indicate the standard error (SE) of the mean. The different lowercase letters indicate significant differences (a, b, and c represent *p* < 0.05).

**Figure 7 ijms-23-12537-f007:**
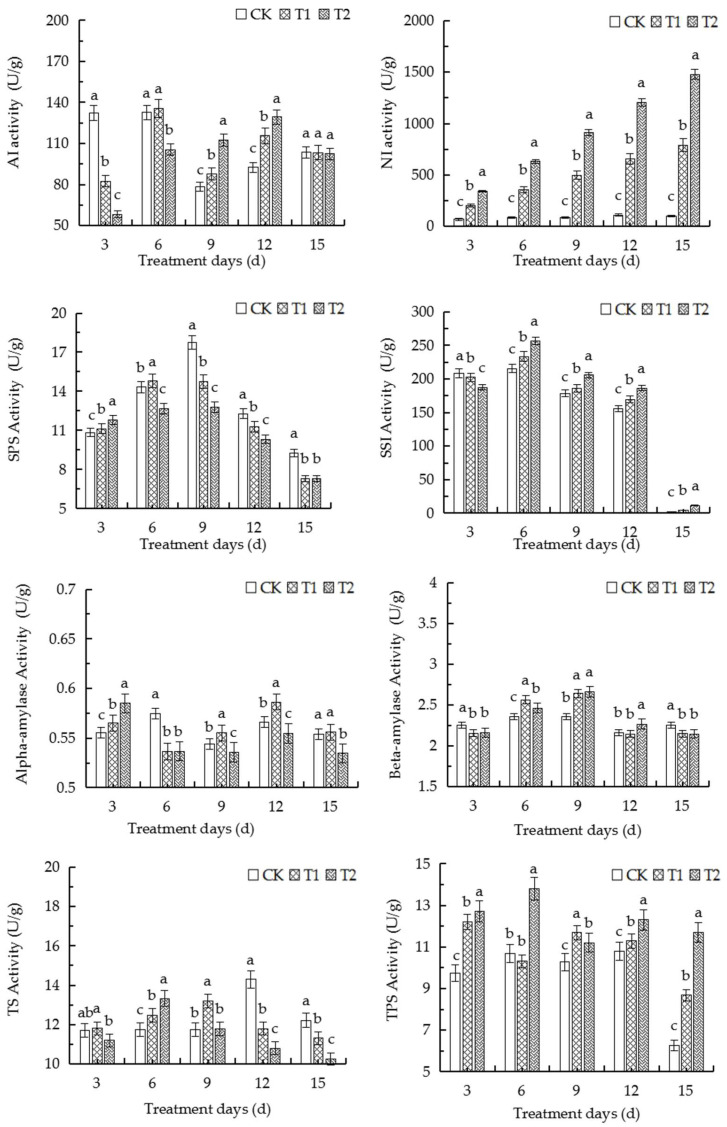
The activity of the enzymes related to sugar and starch metabolism in the roots under elevated root-zone CO_2_, including AI, NI, SPS, SSI, α-amylase, β-amylase, TS, and TPS. The biological replicates were performed in triplicate. The error bars indicate the standard error (SE) of the mean. The different lowercase letters represent significant differences (a, b, and c represent *p* < 0.05).

**Figure 8 ijms-23-12537-f008:**
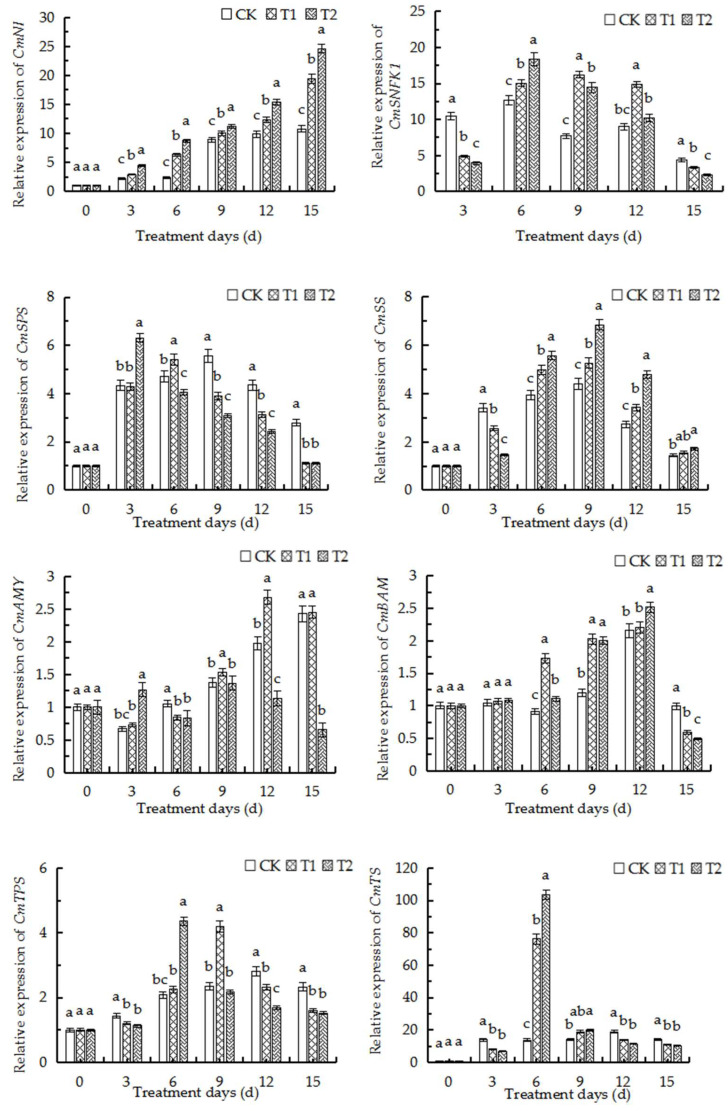
qRT-PCR analysis of major genes related to sugar and starch metabolism. The relative expression of *CmNI, CmSNFK1*, *CmSPS*, *CmSS, CmAMY, CmBAM, CmTPS* and *CmTS* genes in oriental melon seedling roots on the 6th day under elevated root-zone CO_2_ treatments were measured. Significant differences between treatments and control are marked with lowercase letters (a, b and c represent *p* < 0.05).

**Figure 9 ijms-23-12537-f009:**
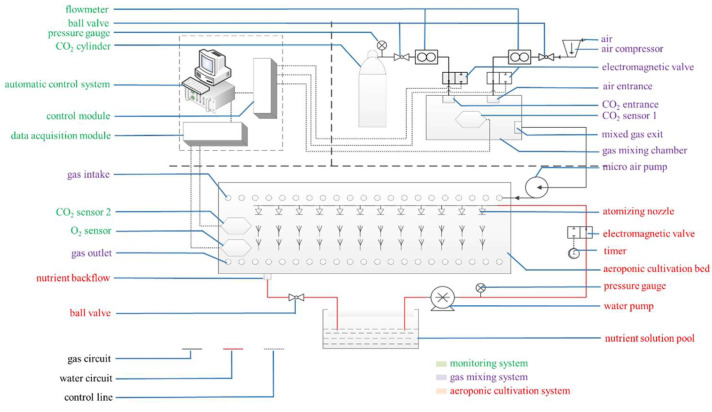
Root-zone CO_2_ cultivation and treatment equipment (quoted Chen et al., 2020 [24]).

**Table 1 ijms-23-12537-t001:** Root growth indices and root vigor at different root-zone concentrations of CO_2_.

Root Growth Indices	Treatment Groups	Days of Treatment (Day)				
		3	6	9	12	15
Total root length (cm)	CK	843.67 ± 1.53 c	1107.33 ± 2.08 c	1476.11 ± 2.21 a	1597 ± 1.32 a	1724.15 ± 2.65 a
	T1	903.21 ± 1.24 b	1175.55 ± 2.65 a	1333.67 ± 3.06 b	1396.67 ± 1.65 b	1522.67 ± 3.06 b
	T2	1099.54 ± 1.21 a	1127.33 ± 2.08 b	1256.33 ± 2.52 c	1322.24 ± 2.02 c	1456.65 ± 2.03 c
Total root surface area (cm^2^)	CK	206.33 ± 2.52 c	466.33 ± 1.53 a	906 ± 1.23 a	1467.67 ± 1.53 a	1564.33 ± 1.35 a
	T1	296.67 ± 3.21 b	462.33 ± 2.58 b	864 ± 3.61 b	1434.33 ± 3.52 b	1493.67 ± 4.04 b
	T2	436 ± 2.65 a	455.67 ± 3.06 b	714.67 ± 2.52 c	1304.33 ± 0.58 c	1363 ± 3.02 c
Total root volume (cm^3^)	CK	0.87 ± 0.03 c	1.33 ± 0.04 b	1.55 ± 0.02 b	2.64 ± 0.02 a	2.93 ± 0.02 a
	T1	1.04 ± 0.01 b	1.37 ± 0.01 b	1.63 ± 0.01 a	2.37 ± 0.02 b	2.37 ± 0.02 b
	T2	1.26 ± 0.02 a	1.42 ± 0.01 a	1.37 ± 0.01 c	1.73 ± 0.02 c	2.05 ± 0.03 c
Root vigor (μg·g^−1^·h^−1^)	CK	12.66 ± 0.14 c	13.67 ± 0.17 c	15.08 ± 0.14 a	5.98 ± 0.16 a	16.45 ± 0.17 a
	T1	13.46 ± 0.12 b	14.86 ± 0.18 b	14.56 ± 0.11 b	15.26 ± 0.12 b	15.76 ± 0.14 b
	T2	14.97 ± 0.12 a	17.58 ± 0.11 a	13.13 ± 0.13 c	13.07 ± 0.12 c	12.99 ± 0.12 c
Total root tip number	CK	867 ± 1.53 c	1163 ± 2.08 c	1446 ± 2.09 a	1543 ± 1.53 a	1685 ± 3.12 a
	T1	1006 ± 1.28 b	1277 ± 1.53 ab	1403 ± 1.04 a	1499 ± 1.35 b	1543 ± 1.53 b
	T2	1222 ± 4.74 a	1424 ± 3.21 a	1295 ± 3.21 c	1323 ± 3.21 c	1312 ± 2.65 c
0–0.5 mm root tip number	CK	255 ± 1.21 c	276 ± 2.08 c	246 ± 1.52 b	265 ± 1.15 a	244 ± 1.53 a
	T1	337 ± 1.73 a	354 ± 1.44 a	274 ± 2.04 a	235 ± 1.97 b	221 ± 1.54 b
	T2	280 ± 1.09 b	310 ± 1.14 b	221 ± 1.11 c	181 ± 0.54 c	158 ± 1.73 c
0.5–2.0 mm root tip number	CK	489 ± 1.21 c	516 ± 2.28 c	667 ± 1.44 a	681 ± 1.71 a	713 ± 1.62 a
	T1	544 ± 2.05 b	551 ± 1.15 b	598 ± 1.61 b	620 ± 1.34 b	657 ± 2.04 b
	T2	590 ± 2.00 a	655 ± 1.25 a	567 ± 0.26 c	580 ± 0.88 c	602 ± 1.53 c

CK: CO_2_ concentration in greenhouse soil (0.2%, control check); T1: elevated root-zone CO_2_ concentration treatment of 0.4%; T2: elevated root-zone CO_2_ concentration treatment of 1.1%. Duncan’s multiple range tests were conducted: Significant differences between control and treatments are marked with lowercase letters (a, b and c represent *p* < 0.05).

**Table 2 ijms-23-12537-t002:** Summary of transcriptome sequencing data.

Sample Name	Clean Reads	Clean Bases	% ≥Q30	Mapped Reads	Unique Mapped Reads	GC Content
CK-1	20,972,383	6,281,865,348	92.95%	39,476,959 (94.12%)	38,780,024 (92.45%)	43.84%
CK-2	21,989,492	6,587,534,178	92.62%	41,278,506 (93.86%)	40,506,664 (92.10%)	44.19%
CK-3	19,324,296	5,782,784,588	92.88%	36,309,424 (93.95%)	35,665,517 (92.28%)	43.79%
T1-1	21,680,097	6,481,877,992	92.94%	40,640,140 (93.73%)	39,880,455 (91.97%)	44.48%
T1-2	22,140,865	6,624,929,878	92.68%	41,229,200 (93.11%)	40,478,885 (91.41%)	44.45%
T1-3	21,089,895	6,312,995,220	92.76%	39,347,813 (93.29%)	38,609,369 (91.54%)	44.37%
T2-1	21,309,551	6,376,761,952	93.05%	39,664,868 (93.07%)	38,978,316 (91.46%)	44.52%
T2-2	21,762,284	6,517,388,230	92.82%	40,803,032 (93.75%)	40,125,589 (92.19%)	43.96%
T2-3	22,392,078	6,709,294,358	92.91%	42,016,764 (93.82%)	45,076,178 (92.90%)	44.39%

Clean Reads: the total number of pair-end reads in Clean Data; Clean Bases: the total number of bases in Clean Data; ≥Q30%: the percentage of bases with a Clean Data quality value of 30 or more. Mapped Reads: reads that are compared to the melon reference genome, which is the sum of all reads for each sample; Unique Mapped Reads: reads that are compared to the unique position of the melon reference genome; GC content: the percentage of G and C bases in Clean Data.

**Table 3 ijms-23-12537-t003:** DEGs related to sugar and starch metabolism.

Comparison between Treatments	Gene ID	Gene Name	FDR	Log_2_Fold Change	Regulation
T1/CK	MELO3C006362.2	*BAM*	1.29 × 10^−^^35^	1.549877695	up
	MELO3C005751.2	*TS*	2.22 × 10^−^^65^	1.073184869	up
	MELO3C026210.2	*SNFK1*	1.76 × 10^−^^45^	1.019037441	up
	MELO3C017002.2	*AMY*	2.21 × 10^−^^7^	−1.237429382	down
T2/CK	MELO3C006362.2	*BAM*	1.39 × 10^−^^17^	1.210496866	up
	MELO3C005751.2	*TS*	0	3.029269697	up
	MELO3C025673.2	*TPS*	5.77 × 10^−^^185^	1.669152326	up
	MELO3C017942.2	*SS*	7.66 × 10^−^^55^	1.17349538	up
	MELO3C026210.2	*SNFK1*	3.87 × 10^−94^	1.44586252	up
	MELO3C006727.2	*NI*	1.02 × 10^−^^55^	1.130833392	up
	MELO3C017002.2	*AMY*	2.21 × 10^−^^7^	−1.237429382	down
T2/T1	MELO3C025673.2	*TPS*	5.77 × 10^−^^185^	1.669152326	up
	MELO3C005751.2	*TS*	0	3.029269697	up
	MELO3C006727.2	*NI*	1.02 × 10^−^^55^	1.130833392	up
	MELO3C020357.2	*SPS*	1.86 × 10^−^^8^	−1.040248354	down

FDR: statistical value of significant difference; Log2Fold Change: logarithmic value of multiple of differential expression; Regulation: relative expression status of gene compared to CK.

**Table 4 ijms-23-12537-t004:** Specific primers used for qRT-PCR.

Gene ID		Sequence (5^′^–3^′^)
Actin	F	AAGGCAAACAGGGAGAAGATGA
	R	AGCAAGGTCGAGACGTAGGATA
MELO3C017002.2	F	CCCAACCTTTTGTCCGTAGAG
	R	CAATCCAATTTATTATCCGCTGT
MELO3C013887.2	F	TGATGATGCCGTTGGACAGT
	R	CCGTGCTTTTTAGCCATTTCT
MELO3C017942.2	F	TTTACTCGTAGCCCCAGCGT
	R	AATGGCGGCAGAACAATAGC
MELO3C020357.2	F	CGTGCTCGGCGTGGAGTA
	R	GGAGAGGACCCATCGCTTGTA
MELO3C005751.2	F	AACAGAAGTCAGCCTCCCCTT
	R	TGATGCGATTTCACGGTATAGAT
MELO3C025673.2	F	ATTGGCGTGATAAAGTTGTTTTG
	R	ATATAGCGCACACAAGTGCATGA
MELO3C026210.2	F	GAAATGCGTCGAAGAAGGCT
	R	TCGTGTTGTCCCAACGGTGT
MELO3C006727.2	F	CTTTATGTCTGTCGGAGGGTT
	R	GCTTCACAATACGCTCTATGCACT

F: Forward. R: Reverse.

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
