# Peer review of "Physiological and Transcriptomic Analyses Reveal the Effects of Elevated Root-Zone CO2 on the Metabolism of Sugars and Starch in the Roots of Oriental Melon Seedlings"

_ijms, 2022, doi:10.3390/ijms232012537_

Round 1
Reviewer 1 Report
The manuscript is well-written but there are a few grammatical aspects that the authors need to correct, and, to help with this, I have included comments in the annotated pdf that I attach to my report.
Also, I have noticed that the authors consistently use "root-zone" instead of "root zone". I think this should be changed to "root zone".
In the microscope images, there is an indication of the scale as 5 micrometers or 1 micrometer in some instances but there is no visible bar/line that indicates the 1 micrometer or the 5 micrometer. This must be included in a visible color that will stand out against the background of the images (I think a white bar/line would work for the black background of the images).
Besides that, I am happy wit the manuscript.
